# Perspectives from designated family caregivers of critically ill adult patients during the COVID-19 pandemic: A qualitative interview study

**Stephana J. Moss**[1,2,3,4], **Karla D. Krewulak**[1,2], **Henry T. Stelfox**[1,2,3,5], **Scott B. Patten**[1,3,4,5,6], **Christopher J. Doig**[1,2,5], **Jeanna Parsons Leigh**[2,6], **Kirsten M. Fiest**[1,2,3,4,7]*

1 Department of Community Health Sciences, Cumming School of Medicine, University of Calgary, Calgary, Alberta, Canada, 2 Department of Critical Care Medicine, University of Calgary and Alberta Health Services, Calgary, Alberta, Canada, 3 O'Brien Institute for Public Health, University of Calgary, Calgary, Alberta, Canada, 4 Hotchkiss Brain Institute, University of Calgary, Calgary, Alberta, Canada, 5 Alberta Health Services, Edmonton, Alberta, Canada, 6 Faculty of Health, School of Health Administration, Dalhousie University, Halifax, Nova Scotia, Canada, 7 Department of Psychiatry, Cumming School of Medicine, University of Calgary, Calgary, Alberta, Canada

* kmfiest@ucalgary.ca

**Data Availability Statement:** Data cannot be shared publicly because of ethical restriction (i.e., patient confidentiality; data contains potentially

## Abstract

### Background

Family visitation in intensive care units (ICU) has been impacted by the severe acute respiratory syndrome coronavirus 2 (COVID-19) pandemic. While studies report on perceptions of families completely restricted from ICUs, little is known about the burden experienced by designated family caregivers allowed to visit their critically ill loved one. This study sought the perspectives of family caregivers of critically ill patients on the impact of one-person designated visitor policies mandated in ICUs during the COVID-19 pandemic.

### Methods

Throughout the study period a restricted visitation policy was mandated capturing the first (April 2020) and second (December 2020) waves of the pandemic that allowed one designated family caregiver (i.e., spouses or adult children) per patient to visit the ICU. Designated family caregivers of critically ill patients admitted to ICU September 2020 to November 2020 took part in individual 60-minute, semi-structured interviews at 6-months after discharge from the index ICU admission. Themes from family interviews were summarized with representative quotations.

### Results

Key themes identified following thematic analysis from six participants included: one visitor rule, patient advocate role, information needs, emotional distress, strategies for coping with challenges, practicing empathy, and appreciation of growth.

sensitive information). Data may be available upon reasonable request from the University of Calgary Conjoint Health Research Ethics Board and Alberta Health Services research and innovation administration (contact via chreb@ucalgary.ca and research.administration@ahs.ca) for researchers who meet the criteria for access to confidential data.

**Funding:** SJM was supported by a Canadian Institutes of Health Research Doctoral Research Award. The funder had no role in study design, data collection and analysis, decision to publish, or preparation of the manuscript; no financial relationships with any organizations that might have an interest in the submitted work in the previous three years; no other relationships or activities that could have influenced the submitted work.

**Competing interests:** The authors have declared that no competing interests exist.

## Conclusion

Designated family caregivers of critically ill patients admitted to ICU during the COVID-19 pandemic perceived a complex and highly stressful experience. Support from ICU family liaisons and psychologists may help ameliorate the impact.

## Introduction

Critically ill patients admitted to the intensive care unit (ICU) are among the sickest patients in the healthcare system given their need for urgent treatment with life sustaining technologies [1]. Family caregivers of critically ill patients experience distress, as witnessing critical illness and intense ICU therapies can elicit feelings of helplessness [2]. Family caregivers frequently experience long-lasting, negative psychological consequences, including anxiety, depression, post-traumatic stress disorder, emotional distress, and sleep disturbances [3].

In response to the burden of critical illness for family caregivers of ICU patients, the Society of Critical Care Medicine Guideline for Patient and Family-Centred Care recommends regular visitation between family caregivers and ICU patients to improve outcomes (e.g., distress) [4, 5] and experiences (e.g., satisfaction) [6, 7] among ICU patients and their families. Most hospitals, including the intensive care units, enacted restricted visitation policies as part of infection control measures [8] to limit spread of the COVID-19 virus, reduce use of personal protective equipment, and to facilitate organizing care [9, 10]. Well-intentioned, restricted visitation policies may have unintended negative consequences on family caregivers, such as grief over inadequate communication and sparse involvement in the provision of care [11, 12]. Designated family caregivers of critically ill patients admitted to ICUs that mandated one-person designated visitor policies faced additional challenges when having to deliver medical information to other family members that were restricted from visiting [13, 14].

Restricted visitation in the ICU during the COVID-19 pandemic may lead to long-term detriment [15]. Perspectives from designated family caregivers of critically ill patients are unknown. The objective of this study was to describe perspectives of designated family caregivers of critically ill patients on the impact of one-person designated visitor policies mandated in ICUs during the COVID-19 pandemic.

## Methods

### Study design

This qualitative study was conducted at Foothills Medical Centre ICU (Calgary, AB, Canada) between September 2020 to November 2020. A restricted visitation policy was mandated throughout the study period (March 2020 to May 2021); capturing the (entire) first and second waves and (part of the) third wave of the pandemic that allowed one designated family caregiver per patient to visit. We used a qualitative descriptive approach [16] with data collected from semi-structured interviews with designated family caregivers (i.e., spouses or adult children designated to visit the ICU routinely) of critically ill patients in accordance with the Consolidated Criteria for Reporting Qualitative Research (COREQ) (S1 Table) [17]. The Conjoint Health Research Ethics Board at the University of Calgary approved this study (Ethics ID: REB19-1000). Informed consent and oral consent were sought from all participants that agreed to be interviewed.

## Selection and description of participants

We used a convenience sample, of designated family caregivers who participated in another (ongoing) study by our group and indicated interest in being contacted to participate in additional research projects [18]. Family caregivers were adults (≥18 years), able to understand and communicate in English, and able to provide informed consent. We invited family caregivers using the contact information they provided (e-mail or telephone).

## Semi-structured interview guide

A multidisciplinary research team (patient partner (B.S.), doctoral student (S.M.), research assistant (I.Y.), research associate (K.K), epidemiologist (K.F.), and qualitative research expert (J.P.L.)) created a draft semi-structured interview guide based on research experience and relevant literature [14, 19, 20]. For feedback and to ensure quality control, draft semi-structured interview guides were presented to a patient partner (M.A., a community member involved with our research team) and their family caregiver (J.A.), as well as a research coordinator (C.G.), all of whom had no prior involvement in the research study. A revised interview guide was then drafted, and pilot tested independently on three occasions in interviews with two critical care nurses (K. W., V.O.), and an intensivist (N.J.). The set interview guide was refined iteratively based on feedback from pilot interviews; no further edits were required after this point (S2 Table).

## Data collection

Demographic data on patients and family caregivers was collected upon enrollment in the larger RCT. Telephone interviews were conducted by S.M. who has experience planning and facilitating semi-structured interviews. Two days prior to each interview, participants (with e-mail access) were sent information about the interview objectives. Participant oral consent was obtained by the research team prior to the start of each interview. All interviews were conducted within 60-minutes, audio recorded, transcribed verbatim, de-identified, and imported to NVivo-12 (QSR International, Melbourne, Australia) for data management.

## Data analysis

We analyzed demographic data by using descriptive statistics. All variables were categorical and reported as counts and proportions. Analysis of qualitative data was conducted concurrently and iteratively using a thematic synthesis approach published by Braun and Clarke [21]. We used a data-driven inductive approach to coding [22] that allowed our working knowledge of the topic [23] to guide the structure of interview discussions while permitting themes to emerge directly from the data [24]. The coding process included two coders (S.M., K.K.) who carefully read all transcripts before coding one-third of the data set to generate initial codes. Once the initial set of codes was developed, coders switched transcripts to ensure that all were coded in duplicate. The two coders searched for themes by collating codes across the data set and met biweekly for one month to refine themes and discuss progress. Two participants were provided with a copy of the final list of themes and sub-themes to review and comment on to ensure credibility, accuracy, and validity. We formally compared themes across participants and compared interpretations across researchers in order to ensure analytic rigor.

# Results

## Participants

Ten designated family caregivers participated in another study by our group from September 2020 to November 2020, of which eight (n = 8, 80%) indicated interest in being contacted to

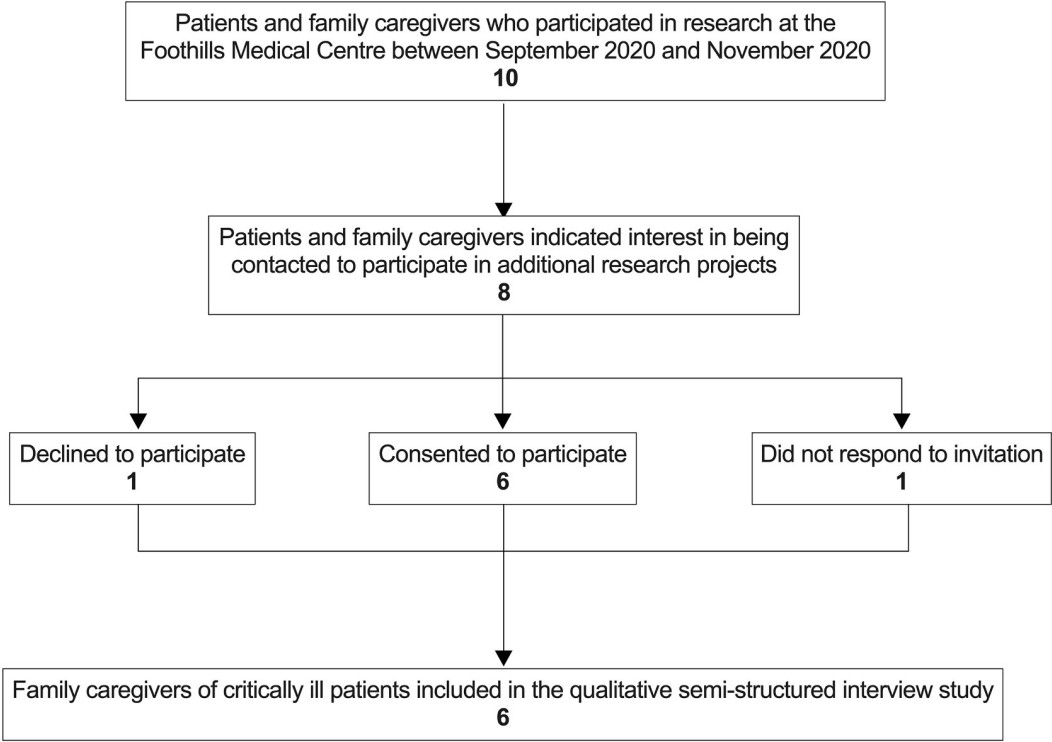

**Fig 1. Flow diagram of selection of family caregivers for interviews.**

participate in additional research projects through a telephone call (*n* = 2, 25%) or an e-mail invitation (*n* = 6, 75%) (Fig 1). Six (75%) family caregivers agreed to participate in the telephone interview. Interviews were conducted at an average of 6.3 months (standard deviation [SD] 2.3) post-ICU discharge.

Family caregivers were mostly female (*n* = 4, 67%), of North American descent (*n* = 4, 67%), and had completed some university/college, without receiving a degree (*n* = 4, 67%). Half (*n* = 3, 50%) of the participants were spouses of critically ill patients (Table 1). Some family caregivers (*n* = 2, 33%) self-reported being diagnosed or treated for depression (prior to ICU admission) and half (*n* = 3, 50%) self-reported being diagnosed or treated for anxiety (prior to ICU admission). Critically ill patients (related to family caregiver participants) were mostly male (*n* = 4, 67%) with either some high school (*n* = 2, 33%) or a Master's degree (*n* = 2, 33%). No patient was diagnosed with the COVID-19 virus prior to or during their ICU stay.

## Family caregiver perceptions

Designated family caregivers of critically ill patients admitted to ICU during the COVID-19 pandemic perceived a complex and highly stressful experience. Support from ICU family liaisons and psychologists may help ameliorate the impact. Participants described their experiences with having to process their loved one's prognosis and treatment information, engage in shared decision making, and then relay information to family members who were not allowed to visit.

Seven themes related to caring for a critically ill patient as the designated family caregiver during the COVID-19 pandemic were identified in the data: (1) one visitor rule (mandated in restricted visitation policies); (2) patient advocate role (being present to weigh in with the

**Table 1. Demographics of designated family caregiver participants and their critically ill loved one admitted to an intensive care unit during the COVID-19 pandemic.**

| Demographic | Family Caregivers (N = 6) | Critically Ill Patients (N = 6) |
|---|---|---|
| *Sex*[1] | | |
| Male | 2 (33.3%) | 4 (66.7%) |
| Female | 4 (66.7%) | 2 (33.3%) |
| *Gender*[2] | | |
| Male | 2 (33.3%) | 4 (66.7%) |
| Female | 4 (66.7%) | 2 (33.3%) |
| *Ethnic or cultural group*[3] | | |
| Other North American | 4 (66.7%) | 3 (50.0%) |
| First Nations | 1 (16.7%) | 0 (00.0%) |
| Eastern European | 2 (33.3%) | 2 (33.3%) |
| Western European | 2 (33.3%) | 1 (16.7%) |
| British Isles | 1 (16.7%) | 0 (00.0%) |
| *Education*[4] | | |
| Some high school | 0 (00.0%) | 2 (33.3%) |
| High school graduate | 0 (00.0%) | 1 (16.7%) |
| Some university/college (no degree) | 4 (66.7%) | 0 (00.0%) |
| Bachelor's degree | 2 (33.3%) | 0 (00.0%) |
| Master's degree | 0 (00.0%) | 2 (33.3%) |
| Professional degree | 0 (00.0%) | 1 (16.7%) |
| *Relationship to patient* | | |
| Spouse or Common-Law | 3 (50.0%) | - - |
| Adult Child | 3 (50.0%) | - - |
| *Considered primary caregiver*[5] | | |
| Yes | 6 (100.0%) | - - |
| No | 0 (00.0%) | - - |
| *Clinically relevant depression*[6] | | |
| Yes | 2 (33.3%) | - - |
| No | 4 (66.7%) | - - |
| *Clinically relevant anxiety*[6] | | |
| Yes | 3 (50.0%) | - - |
| No | 3 (50.0%) | - - |

Numbers are counts with percentages. Dashes indicate that the question was not asked.

[1] Recorded at birth

[2] Reported as gender identity

[3] Multiple selections per participant were allowed

[4] Highest degree received

[5] During patient stay in the ICU and after discharge from hospital

[6] Relating to direct medical treatment by a healthcare professional prior to ICU admission as self-reported by the participant

clinical care team); (3) information needs (receiving regular and clear information); (4) emotional distress (toward their critically ill loved one); (5) strategies for coping with challenges (related to restricted visitation policies); (6) practicing empathy (with members of the ICU care team); and (7) appreciation of growth (despite hardships) (Table 2).

 **One visitor rule.** Participants from all interviews provided their perspectives on the challenge of adhering to the one designated visitor policy. Difficult for all, this policy was especially

**Table 2. Perspectives of designated family caregivers on caring for critically ill loved ones admitted to an intensive care unit during the COVID-19 pandemic.**

| Themes | Quotes |
|---|---|
| **One Visitor Rule** | *At that time, one visitor was allowed that had to be the same visitor. So, our children were not allowed to visit, which was really hard on them all. Okay, we thought, we have to deal with this. (Spouse)*<br>*I just couldn't imagine him being there all by himself—I just had to be there, it was very important to me. (Son)*<br>*It was absolutely harder because of COVID. I felt like I had to be there all of the time because only one person was allowed, and I didn't ever want him to feel like he was alone. He didn't understand why the kids couldn't be there—that made it very hard, not being able to have that support. (Spouse)* |
| **Patient Advocate Role** | *I was there and I saw the lung exercises, so then I could quiz him later on. I asked, did you do it, do you remember how long you're supposed to do it? You're supposed to do it every hour. Are you doing it? Those people out there really care, and I want you to do your lung exercise. But what I know is that is that was just a phone call [if no visitors were allowed in the ICU]l, I wouldn't know those specifics and I wouldn't be able to watch and be an advocate. (Spouse)*<br>*Even though I was the only one in there I was never asked to pipe up, to tell them about what he is really like, to advocate—I didn't know how you know, being alone. (Spouse)*<br>*The lessons learned is I wish I would have been more involved in rounding. I wish I would've been more invited. I know it's hard right now, given [COVID-19] restrictions, but if I was even just listening, I would have felt included. (Daughter)* |
| **Information Needs** | *I'm a very curious person and I like to know what's going on. So, I spent a long time asking questions whenever they were there. The staff was very good at giving us answers, but yet a lot of the time we were waiting—sitting and waiting or going back and forth and waiting for them to come to your patient. (Daughter)*<br>*I was afraid to ask questions. I felt isolated. Not a lot of information was given. I felt intimidated. (Spouse)*<br>*I had to give daily updates to everybody. I would wait for a report and then I would go out to call everybody, let them know how he was doing. So many calls—I had to make myself a pretty decent schedule. The only things I had time for were to come in [to the ICU], make all my calls, go back to the hotel, turn on TV for a few minutes and then go to bed. (Spouse)* |
| **Emotional Distress** | *It was an emotional and very difficult experience. I felt scared, left out, kind of anxious. (Daughter)*<br>*I spent many hours just sitting there wondering, you know, listening to machines, beeping, very loudly. I was on an emotional rollercoaster—I couldn't find solitude. (Spouse)*<br>*I think that you fool yourself into thinking that you're okay. I'm okay. I'm okay. I'm okay, I kept saying. You just are running on adrenaline, right. I didn't relax until he left ICU and I know it's the times, right. I had to tell myself that it was fine. . .initially anyway. (Spouse)* |
| **Strategies for Coping with Challenges** | *I knew my lifestyle. I had to eat better, you know, as far as getting some sleep at night. And so, sometimes I wouldn't come back a second time, and there was one time I remember I actually felt really guilty. (Son)*<br>*I'm very scheduled. So, I made myself a daily schedule. Mostly for my own mental health. (Son)*<br>*I made myself a decent schedule. . . day in and day out. That helped for all of us. (Spouse)*<br>*I wrote everything down. I would write it down, what the care team said for the entire day, and then at end of each day I would write everything in a second book that I left for my husband so that he could look back and find what he needed to know, even when I wasn't there. (Spouse)* |
| **Practicing Empathy** | *Being able to talk to them [the ICU care team] provided a feeling of solidarity. They were going through a hard time. You know, you do take an interest in other people who are always there. The talking helped, like teamwork. (Son)*<br>*I would suggest a [virtual] peer group of people that have to deal with these issues, so we can exchange coping mechanisms and ideas, and show empathy for others, you know, that would be helpful. (Spouse)* |

*(Continued)*

**Table 2.** (Continued)

| Themes | Quotes |
|---|---|
| **Appreciation of Growth** | *I mean, once you've been through these stressful, traumatic, draining situations, you look back and reflect and think, great, I can do tough things because I've dealt with a lot. (Spouse)*<br>*I'm very happy that it's over. It gets better and I've learned a lot—I've grown and hey, that's not what I was expecting to say. (Son)* |

burdensome to families with young children: "At that time, one visitor was allowed that had to be the same visitor. So, our children were not allowed to visit, which was really hard on them all." (Spouse). Most designated family caregivers agreed about the guilt when absent from the unit: "I just couldn't imagine him being there all by himself—I just had to be there, it was very important to me" (Adult Child). One family caregiver remarked that not having external support (present with them in the ICU) was challenging:

> It was absolutely harder because of COVID. I felt like I had to be there all of the time because only one person was allowed, and I didn't ever want him to feel like he was alone. He didn't understand why the kids couldn't be there—that made it very hard, not being able to have that support.
>
> (Spouse)

**Patient advocate role.** All designated family caregivers shared their perspectives on the importance of being present that provided an opportunity for the family caregiver (who knows the patient best) to weigh in on subtleties they may notice in the patient's overall demeanor. Family caregivers took opportunities to be actively involved in care of their loved one:

> I was there and I saw the breathing exercises, so then I could quiz him later on. I asked, did you do it, do you remember how long you're supposed to do it? You're supposed to do it every hour. Are you doing it? Those people out there really care, and I want you to do your breathing exercise. But what I know is that if that was just a phone call [if no visitors were allowed in the ICU], I wouldn't know those specifics and I wouldn't be able to watch and be an advocate.
>
> (Spouse)

In contrast, some family caregivers described feeling distress about being involved in patient care and were waiting to be asked to weigh in. One family caregiver remarked: "Even though I was the only one there I was never asked to pipe up, to tell them about what he is really like, to advocate—I didn't know how you know, being alone" (Spouse). The lack of invitation evoked feelings of isolation in one family caregiver: "The lesson learned is I wish I would have been more involved in rounding. I wish I would've been more invited. I know it's hard right now, given [COVID-19] restrictions, but if I was even just listening, I would have felt included" (Adult Child).

**Information needs.** Participants shared their need to receive regular and clear information from the healthcare team regarding their loved one. One family caregiver began to ask questions in order to feel more involved: "I'm a very curious person and I like to know what's going on. So, I spent a long time asking questions whenever they were there. The staff was very

good at giving us answers, but yet a lot of the time we were waiting—sitting and waiting or going back and forth and waiting for them to come to your patient" (Adult Child). However, despite being present, designated family caregivers felt absent without the support of their family in the ICU. A spouse pronounced: "I was afraid to ask questions. I felt isolated. Not a lot of information was given. I felt intimidated" (Spouse). In addition, participants unanimously described the burden of having to relay medical information to remaining family members who were not allowed to visit in the ICU. The significance of other family members in the ICU was conspicuous by their absence:

> *I had to give daily updates to everybody. I would wait for a report and then I would go out to call everybody, let them know how he was doing. So many calls—I had to make myself a pretty decent schedule. The only things I had time for were to come in [to the ICU], make all my calls, go back to the hotel, turn on TV for a few minutes and then go to bed.*
>
> *(Spouse)*

**Emotional distress.**    Designated family caregivers described feeling emotionally distressed for their critically ill loved one. One family caregiver recalled: "It was an emotional and very difficult experience. I felt scared, left out, kind of anxious" (Adult Child). Families were mindful of the unnatural and lonely feeling of being in the ICU without other family members: "I spent many hours just sitting there wondering, you know, listening to machines, beeping, very loudly. I was on an emotional rollercoaster—I couldn't find solitude" (Spouse). Caring for a critically ill patient without support of other family, during a pandemic, sometimes involved self-affirmations:

> *I think that you fool yourself into thinking that you're okay. I'm okay. I'm okay. I'm okay, I kept saying. You just are running on adrenaline. Right? I didn't relax until he left ICU and I know it's the times, right. I had to tell myself that it was fine. . .initially anyway.*
>
> (Spouse)

**Strategies for coping with challenges.**    "Mostly for my own mental health" (Adult Child), while others echoed: "I made myself a decent schedule. . . day in and day out. That helped for all of us" (Spouse). When hospitalized, separated, and isolated at night, one spouse recounted their strategy to cope with the challenge of restricted visitation:

> *I wrote everything down. I would write it down, what the care team said for the entire day, and then at end of each day I would write everything in a second book that I left for my husband so that he could look back and find what he needed to know, even when I wasn't there.*
>
> (Spouse)

**Practicing empathy.**    Participants described practicing empathy with members of the ICU care team, rather than other family caregivers, as waiting rooms were closed and caregivers from different families were not allowed to interact in the ICU. Bearing witness to the challenges faced by other family caregivers, one adult child shared: "Being able to talk to them [the ICU care team] provided a feeling of solidarity. They were going through a hard time. The talking helped, like teamwork" (Adult Child). Families recommended potential avenues for

designated family caregivers to provide support to each other throughout restricted visitation: "I would suggest a [virtual] peer group of people that have to deal with these issues, so we can exchange coping mechanisms and ideas, and show empathy for others, you know, that would be helpful" (Spouse).

**Appreciation of growth.**   All designated family caregivers who participated shared their perspectives of the negative impact of the pandemic and shared lessons learned from providing care:

*I mean, once you've been through these stressful, traumatic, draining situations, you look back and reflect and think, great, I can do tough things because I've dealt with a lot.*

(Spouse)

In the end, designated family caregivers, tired and isolated, described the influence of being resilient on their own, personal growth: "I'm very happy that it's over. It gets better and I've learned a lot—I've grown and hey, that's not what I was expecting to say" (Adult Child).

## Discussion

We conducted a semi-structured interview study to explore perspectives of family caregivers of critically ill patients on the impact of one-person designated visitor policies mandated in ICUs during the COVID-19 pandemic. Our findings indicated that practices to control spread of the SARS-CoV-2 virus changed visitation in the ICU, which transformed the way family caregivers cared for their critically ill loved one. In the context of one-person designated visitor policies in the ICU, these changes led to complex situations that had communication and emotional consequences for family caregivers. The unintended repercussions experienced by designated family caregivers largely hinged on the notion that despite being physically present, designated family caregivers felt helpless and isolated from the ICU care team, and guilt related to being the only family member allowed to visit.

Supporting family caregivers is fundamental to the practice of critical care medicine [6] that is rarely easy [25] and has been more challenging in the COVID-19 pandemic [26]. Even with uninterrupted bedside access and idyllic support, family caregivers have high risk of long-term physical and mental health problems [27]. A one designated visitor policy at our institution that was similar to mandated policies at other Canadian [28, 29] and American [30, 31] institutions meant that if two family members were present when their loved one was admitted, they were forced to choose: who will sit alone, vigil, at the bedside, and who will walk away, leaving their critically ill family member and grieving partner behind?

The COVID-19 pandemic resulted in limitations on family caregiver engagement in the ICU and participation in care that completely reengineered their methods to cope and had potential implications on their well-being [32–34]. The issue is that public health, without an understanding of ICU care, broadly directed hospital restrictions usually without an understanding of potential adverse impact and without input and/or feedback from healthcare providers [35]. The evidence that these interventions mitigated spread of the virus (their benefit) was never measured compared to the negative impact to patients and families against which they were applied (the harm) [15, 36–37]. The data suggests that there was harm, and that this should be considered for future pandemic planning which needs to include perspectives from family caregivers on how to best mitigate the negative effects of restricted visitation [38]. Most research has reported on short-term impacts of restricted visitation policies, few including perspectives from family caregivers themselves, and longer-term consequences of restricted visitation policies are vastly understudied [39–41]. In particular, experiences of family caregivers

forced to decide between visiting their loved one or to place themselves at risk of infection from COVID-19 before the understanding of COVID and availability of vaccination. Added care for ICU family caregivers that emphasizes respect, dignity, and humanization, might come in the form of long-term support plans delivered by ICU family liaison teams [42–44] and psychologists [45–47] that may help to ameliorate the impact of the COVID-19 pandemic on designated family caregivers of critically ill patients.

Prior to mandated restricted visitation policies, studies report families being offered an increasingly active role in the ICU in the participation in patient care (e.g., hygiene, orientation, mobility) [48, 49]. This is related to a growing awareness that family caregivers of ICU patients have specific needs including information to understand the diagnosis, prognosis, and treatment in the patient [50], and support in dealing with psychological distress [51]. Participation in care helps to provide families with a feeling of closeness to the patient [52] that may facilitate their sensemaking about the critical illness [53], thus alleviating their stress [54]. Performing some patient care usually left to healthcare professionals may help families to understand the caring nature of ICU treatments, which may otherwise seem highly invasive [55]. Family participation in care can also play a role in decreasing feelings of powerlessness [56] and contribute to a sense of usefulness that may help to alleviate negative mental health consequences such as guilt, grief, or burden [57]. Furthermore, a role in patient care for families may help the care team to emphasize that families are not just visitors but welcome and appreciated members in the ICU [58]. Participation in care is associated with better satisfaction among family caregivers [59]; thus, it is suggested that healthcare professionals should consider encouraging family caregivers (who wish to do so) to participate in patient care with the support of the ICU care team [52, 60, 61].

The strengths of this study include that the interview guide was informed by narratives reported in the COVID-19 pandemic [62–65], co-designed with researchers, patients, and clinicians, and tested in a pilot study with critical care nurses and intensivists. Interviews were conducted individually and at length, which allowed caregiver's time and space to share perspectives to offer important insights on the psychological burden that afflicts designated family caregivers. There are limitations to consider when interpreting the findings of our study. First, the number of participants included in this study was dependent on the interest of family caregivers in being contacted to participate in additional research projects; other studies were paused many times to conserve resources (i.e., personal protective equipment, staff) to combat the COVID-19 pandemic, which limited recruitment. We did not assess non-designated family members and it is possible that important perspectives were missed. Second, we chose a 6-month follow-up as we were cautious about grief experienced by family caregivers who lost loved ones to critical illness [66, 67]. Third, this is a single-centre qualitative study including six family caregivers that may not be transportable to other ICU settings. Additional interviews to collect data past code saturation in order to assess meaning saturation are required for transferability of our results [68]. Finally, our small sampling frame did not achieve adequate representation of sex, gender, education, and socioeconomic status and we were not able to explore sociocultural factors, including cognitive and linguistic barriers, which might impact communication [69, 70].

## Conclusions

Designated family caregivers of critically ill patients in the ICU perceived that restricted visitation policies mandated to control COVID-19 had unintended negative repercussions. The one-person designated visitor policy meant that the designated family caregiver had to process their loves one's prognosis and treatment, make life-changing decisions, and then relay this

information to remaining family in addition to coping with their own concerns. Long-term support plans for family caregivers of critically ill patients delivered by ICU family liaison teams and psychologists may help to ameliorate the impact of the COVID-19 pandemic on designated family caregivers of critically ill patients. Further research with larger and more diverse sample sizes are required to validate our findings from this hypothesis-generating work.

## Supporting information

**S1 Table. Consolidated criteria for reporting qualitative studies (COREQ).**
(DOCX)

**S2 Table. Interview guide.**
(DOCX)

## Acknowledgments

The authors are grateful for the privilege of speaking with families during the ongoing pandemic. The authors would like to acknowledge the individuals who participated in developing (Bonnie Sept, Israt Yasmeen), reviewing (Melanie C. & Jillian Anglin, Chloe M. de Grood), and piloting the interview guide (Krista Wollny, Victoria Owen, Natalia Jaworska).

## Author Contributions

**Conceptualization:** Stephana J. Moss, Karla D. Krewulak, Henry T. Stelfox, Scott B. Patten, Christopher J. Doig, Kirsten M. Fiest.

**Data curation:** Stephana J. Moss.

**Formal analysis:** Stephana J. Moss, Karla D. Krewulak.

**Funding acquisition:** Stephana J. Moss, Kirsten M. Fiest.

**Investigation:** Stephana J. Moss, Kirsten M. Fiest.

**Methodology:** Stephana J. Moss, Karla D. Krewulak, Henry T. Stelfox, Scott B. Patten, Jeanna Parsons Leigh.

**Project administration:** Stephana J. Moss.

**Resources:** Kirsten M. Fiest.

**Software:** Kirsten M. Fiest.

**Supervision:** Henry T. Stelfox, Scott B. Patten, Christopher J. Doig, Jeanna Parsons Leigh, Kirsten M. Fiest.

**Validation:** Stephana J. Moss, Karla D. Krewulak, Scott B. Patten, Christopher J. Doig, Jeanna Parsons Leigh, Kirsten M. Fiest.

**Visualization:** Stephana J. Moss, Karla D. Krewulak.

**Writing – original draft:** Stephana J. Moss, Karla D. Krewulak, Kirsten M. Fiest.

**Writing – review & editing:** Stephana J. Moss, Karla D. Krewulak, Henry T. Stelfox, Scott B. Patten, Christopher J. Doig, Jeanna Parsons Leigh, Kirsten M. Fiest.

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
