## [Decision Letter · Decision Letter 0]

5 Apr 2022

PONE-D-21-35872Perspectives from Designated Family Caregivers of Critically Ill Adult Patients During the COVID-19 Pandemic: A Qualitative Interview StudyPLOS ONE

Dear Dr. Fiest,

Thank you for submitting your manuscript to PLOS ONE. After careful consideration, we feel that it has merit but does not fully meet PLOS ONE’s publication criteria as it currently stands. Therefore, we invite you to submit a revised version of the manuscript that addresses the points raised during the review process.

I encourage you to follow all the changes recommended by the two reviewers.

We look forward to receiving your revised manuscript.

Kind regards,

Marie-Pascale Pomey

Academic Editor

PLOS ONE

Journal Requirements:

Reviewers' comments:

Reviewer's Responses to Questions

**Comments to the Author**

1. Is the manuscript technically sound, and do the data support the conclusions?

Reviewer #1: No

Reviewer #2: Partly

2. Has the statistical analysis been performed appropriately and rigorously? 

Reviewer #1: N/A

Reviewer #2: I Don't Know

3. Have the authors made all data underlying the findings in their manuscript fully available?

Reviewer #1: No

Reviewer #2: Yes

4. Is the manuscript presented in an intelligible fashion and written in standard English?

Reviewer #1: Yes

Reviewer #2: Yes

5. Review Comments to the Author

Reviewer #1: Overall Comments

- This was an interesting and well written piece. However, more work is required, particularly at the level of description and interpretation of the results. See specific comments below.

Reviewer #2: See comments in document. Need information about the main study.

In general, The manuscript described a specific clinical event and social impact. It is a novelty in science. The small sample sizes gave some data to discuss but the conclusion needs to be adresses differently . Experiments seem to have been conducted rigorously, but details in need it in methodology and more information regarding the main research is need it. The conclusions must be drawn appropriately based on the data presented.

6. PLOS authors have the option to publish the peer review history of their article (what does this mean?). If published, this will include your full peer review and any attached files.

Reviewer #1: **Yes: **Bertrand Lebouché MD, PhD

Associate Professor, Department of Family Medicine, Faculty of Medicine and Health Sciences, McGill University

Anish K. Arora, PhD(c), MSc, BSc (Hons)

PhD Candidate, Family Medicine & Primary Care, McGill University, Vanier Scholar, Canadian Institutes of Health Research

Reviewer #2: No

---

## [Author Response · Author response to Decision Letter 0]

8 Sep 2022

September 8, 2022

Emily Chenette, PhD

Editor-in-Chief, PLOS ONE

Dear Dr. Chenette:

Re: Revised Submission PONE-D-21-35872

Thank you for reviewing our paper entitled “Perspectives from Designated Family Caregivers of Critically Ill Adult Patients During the COVID-19 Pandemic: A Qualitative Interview Study” and for inviting us to revise and resubmit. In response to the reviewers’ feedback, we revised the paper and believe we have improved the overall quality and applicability of our original research paper. 

In this Response Letter, we provide an item-by-item response to comments from all Reviewers, and the exact location of each revision (Section, Page, Paragraph) in the new (revised) manuscript. All changes made to the text of the manuscript are shown in yellow highlight. Each comment (verbatim) by the Reviewer is followed by our detailed response, with any relevant text changes provided in quotations. 

Thank you for considering our manuscript for publication. It is not under consideration for publication elsewhere, nor has it been presented in any form; we look forward to your decision.

Yours sincerely,

Kirsten Fiest, PhD

Associate Professor, Departments of Critical Care Medicine, Community Health Sciences, & Psychiatry

Cumming School of Medicine, University of Calgary 

Director, Research & Innovation, Department of Critical Care Medicine, Alberta Health Services

Journal Requirements

RESPONSE: Thank you for providing PLOS ONE’s style requirements that were followed closely in preparing our revised manuscript and associated files. Please do not hesitate to contact us if there are additional requirements that were missed. 

RESPONSE: Thank you for providing the information required for our Data Availability statement. Below is our statement that is also provided on page 19 of our revised manuscript:

Data cannot be shared publicly because of ethical restriction (i.e., patient confidentiality; data contains potentially sensitive information). Data may be available upon reasonable request from the University of Calgary Conjoint Health Research Ethics Board and Alberta Health Services research and innovation administration (contact via chreb@ucalgary.ca and research. administration@ahs.ca) for researchers who meet the criteria for access to confidential data.

Reviewer #2: See comments in document. Need information about the main study.

In general, The manuscript described a specific clinical event and social impact. It is a novelty in science. The small sample sizes gave some data to discuss but the conclusion needs to be adresses differently . Experiments seem to have been conducted rigorously, but details in need it in methodology and more information regarding the main research is need it. The conclusions must be drawn appropriately based on the data presented.

1. Abstract: Date? Period?

RESPONSE: We appreciate that additional information is required to describe when the first and second waves occurred in geographical location. This information has been added to our Abstract (page 9) with the referenced statement that now reads as follows: 

“Throughout the study period a restricted visitation policy was mandated capturing the first (April 2020) and second (December 2020) waves of the pandemic that allowed one designated family caregiver (i.e., spouses or adult children) per patient to visit the ICU.”

2. Data Collection: How was selected the study population? How many patients was asked to participate before reaching 6 people?

RESPONSE: Thank you for your inquiries. We have provided information on how the study population was selected on page 4 of our revised manuscript. We have clarified that the original study is currently ongoing and have provided the reference to the published protocol for the ongoing, original study. 

“We used a convenience sample, of designated family caregivers who participated in another (ongoing) study by our group and indicated interest in being contacted to participate in additional research projects [18].”

Information on how many individuals were asked to participate is provided on page 6 of our revised manuscript that reads as follows:

“Ten designated family caregivers participated in another study by our group from September 2020 to November 2020, of which eight (n=8, 80%) indicated interest in being contacted to participate in additional research projects through a telephone call (n=2, 25%) or an e-mail invitation (n=6, 75%) (Figure 1).” 

3. Participants: These are data describing the patient, not the study population. To clarify

RESPONSE: Thank you for your suggestion to clarify the presentation of our demographic information. The revised sentence on page 7 of our manuscript reads as follows: 

“Critically ill patients (related to family caregiver participants) were mostly male (n=4, 67%) with either some high school (n=2, 33%) or a Master’s degree (n=2, 33%).”

4. Family Caregiver Perceptions: Confused, to be explain

RESPONSE: Thank you for your comment that indicated that perhaps a short description after each of the presented themes would be helpful to situate the reader to our findings. On page 9 of our revised manuscript, we have added brief descriptions after each of the six themes, including the ‘one visitor rule’ that was highlighted in your comment. This sentence now reads as follows:

“Seven themes related to caring for a critically ill patient as the designated family caregiver during the COVID-19 pandemic were identified in the data: (1) one visitor rule (mandated in restricted visitation policies); (2) patient advocate role (being present to weigh in with the clinical care team); (3) information needs (receiving regular and clear information); (4) emotional distress (toward their critically ill loved one); (5) strategies for coping with challenges (related to restricted visitation policies); (6) practicing empathy (with members of the ICU care team); and (7) appreciation of growth (despite hardships) (Table 2).”

5. Discussion: It will important to discuss the fear of caregivers of having COVID vs going in Hospital, particularly before the Understanding of COVID and introduction of vaccination. Risk vs benefits

RESPONSE: We agree with the reviewer that this is an important information that should be included in our discussion. We have added the following statements to page 16 of our revised manuscript to highlight the dearth of information regarding experiences and perspectives of family caregivers in COVID-19, particularly when forced to choose between visiting their loved one or risk of viral infection.

“Most research has reported on short-term impacts of restricted visitation policies, few including perspectives from family caregivers themselves, and longer-term consequences of restricted visitation policies are vastly understudied [40-42]. In particular, experiences of family caregivers forced to decide between visiting their loved one or to place themselves at risk of infection from COVID-19 before the understanding of COVID and availability of vaccination.”

---

## [Editor Report · Decision Letter 1]

15 Sep 2022

Perspectives from Designated Family Caregivers of Critically Ill Adult Patients During the COVID-19 Pandemic: A Qualitative Interview Study

PONE-D-21-35872R1

Dear Dr. Fiest,

We’re pleased to inform you that your manuscript has been judged scientifically suitable for publication and will be formally accepted for publication once it meets all outstanding technical requirements.

Kind regards,

Marie-Pascale Pomey

Academic Editor

PLOS ONE

---

## [Editor Report · Acceptance letter]

19 Sep 2022

PONE-D-21-35872R1 

Perspectives from Designated Family Caregivers of Critically Ill Adult Patients During the COVID-19 Pandemic: A Qualitative Interview Study 

Dear Dr. Fiest:

I'm pleased to inform you that your manuscript has been deemed suitable for publication in PLOS ONE. Congratulations! Your manuscript is now with our production department. 

Kind regards, 

on behalf of

Dr. Marie-Pascale Pomey 

Academic Editor

PLOS ONE